# Quercetin as a Natural Therapeutic Candidate for the Treatment of Influenza Virus

**DOI:** 10.3390/biom11010010

**Published:** 2020-12-24

**Authors:** Parvaneh Mehrbod, Dorota Hudy, Divine Shyntum, Jarosław Markowski, Marek J. Łos, Saeid Ghavami

**Affiliations:** 1Influenza and Respiratory Viruses Department, Pasteur Institute of Iran, Tehran 1316943551, Iran; 2Department of Laryngology, Faculty of Health Sciences in Katowice, Medical University of Silesia, 40-027 Katowice, Poland; dorota@hudy.pl (D.H.); jmarkow1@poczta.onet.pl (J.M.); 3Biotechnology Center, Silesian University of Technology, 44-100 Gliwice, Poland; dshyntum@gmail.com; 4Department of Pathology, Pomeranian Medical University, 71-344 Szczecin, Poland; mjelos@gmail.com; 5Department of Human Anatomy and Cell Science, Rady Faculty of Health Sciences, Max Rady College of Medicine, University of Manitoba, Winnipeg, MB R3E 0J9, Canada; saeid.ghavami@umanitoba.ca

**Keywords:** flavonols, influenza virus, medicinal plants, quercetin, quercetin derivatives, SASRS-COV-2

## Abstract

The medical burden caused by respiratory manifestations of influenza virus (IV) outbreak as an infectious respiratory disease is so great that governments in both developed and developing countries have allocated significant national budget toward the development of strategies for prevention, control, and treatment of this infection, which is seemingly common and treatable, but can be deadly. Frequent mutations in its genome structure often result in resistance to standard medications. Thus, new generations of treatments are critical to combat this ever-evolving infection. Plant materials and active compounds have been tested for many years, including, more recently, active compounds like flavonoids. Quercetin is a compound belonging to the flavonols class and has shown therapeutic effects against influenza virus. The focus of this review includes viral pathogenesis as well as the application of quercetin and its derivatives as a complementary therapy in controlling influenza and its related symptoms based on the targets. We also touch on the potential of this class of compounds for treatment of SARS-COV-2, the cause of new pandemic.

## 1. Introduction

### 1.1. Influenza Virus

Influenza A virus (IAV) from the family Orthomyxoviridae is understood to be the underlying cause for human and animal flu infections [1]. It has an eight-segment negative-sense, single-stranded RNA genome, which codes several functional, structural, and nonstructural proteins, including PB1, PB2, PA, HA, NA, NP, M (1,2), and NS (1,2) (Figure 1), each one of which has a critical role in the function of the virus [2]. Several antigenic drifts and shifts in its genetic structure, especially in HA and NA, have created different types and subtypes of this virus. Among them, three HA (H1, H2, and H3) and two NA (N1 and N2) subtypes have been responsible for human epidemics and pandemics for hundreds of years [3]. These viral proteins play essential roles in viral life cycle steps including attachment and entry, synthesis of viral RNA and proteins, packaging, budding, and release [2].

Influenza afflicts about 5–10% of adults and 20–30% of children annually [4]. Severe disease could result in the deaths of about 290,000–650,000 people worldwide in one season [5]. Beside deaths, influenza may cause respiratory, diabetic, cardiovascular, renal, and neurological complications as reviewed in [6,7]. Additionally, the estimated cost of influenza infection in the United States alone amounts to about $11.2 billion annually [8]. In 2019/2020, the “influenza-season” coincided with the COVID-19 pandemic outbreak and the current (2020/2021) “influenza-season” is coinciding with the peak of COVID-19 epidemics in many countries, increasing the risk of coinfection with both viruses. To date, at least 13 cases of coinfection of Covid-19 and influenza have been reported in PubMed-listed articles. A larger prevalence of coinfection is suspected, though still under investigation due to the symptoms similarities, as reviewed by H. Khorramdelazad et al. (2020) [9]. In some of the reviewed cases, the coinfection was not more severe than infection with COVID-19 alone [10]. One report suggested a protective effect of influenza infection against infection with COVID-19 [11]. Coinfection was also investigated in an animal model study using golden Syrian hamsters [12]. The study results indicated more “clinically” severe disease in the case of coinfection with both viruses. In September 2020, a British SAGE (Scientific Advisory Group for Emergencies) report was released concerning coinfection of COVID-19 and influenza in hospitalized patients. About 18% of hospitalized Covid-19 patients that were tested for influenza were also carrying influenza virus. Only 3.9% of all COVID-19-positive cases were also cross-tested for influenza, and only 0.7% of them were influenza positive [13]. The consequences of coinfections were an increased number of critical care admissions, longer hospitalization time (greater than twice the time), and a greater need for oxygen and mechanical ventilation [13]. As both diseases produce similar symptoms, it is highly recommended to use all possible means to prevent the infection with influenza, as the COVID-19 vaccine is still not widely available.

Yearly vaccination is the primary plan to control this infection; however, annual vaccination has limitations, including the time needed to design and produce the vaccine, high cost and inadequate protection due to antigenic shifts and drifts, dependence on egg-based production, regulatory approval procedures, limited worldwide availability, limited efficacy in elderly and unprimed populations, as well as lack of cross-reactivity by current vaccines [14,15].

Current therapeutic strategies are primarily used for prophylaxis and treatment. One category of anti-influenza drugs, neuraminidase inhibitors (NAI), inhibits influenza virus neuraminidase reducing viral shedding within the respiratory tract [16]. Oseltamivir and zanamivir have been approved in many countries, while peramivir and laninamivir are approved in Japan, and peramivir is approved in China and Korea [17]. Another category, which includes M2 channel blockers amantadine and rimantadine, functions by blocking the viral RNA uncoating within infected cells and preventing its replication by disrupting the function of the transmembrane domain of the viral M2 protein [16].

Severe cases of influenza infection are associated with the risk of death, especially in children, the elderly, and in immunocompromised patients [18]. Despite the improved development of conventional antiviral agents, their effectiveness against influenza viruses is limited [19] and their clinical efficacy is also ambiguous [20,21,22]. Moreover, dramatic increases in resistance of influenza virus against these drugs have been reported [23,24,25,26,27], although resistance to neuraminidase inhibitors has remained at low levels [28]. Severe cases of influenza infection are associated with the risk of death, especially in children, the elderly, and in immunocompromised patients [18]. Therefore, efficient control of influenza outbreaks requires the discovery and development of novel antiviral drugs affecting different aspects of the viral infection pathway. For this purpose, nature might be promising as a reference for targetable pathways and modes.

### 1.2. Viral Pathogenesis

The interactions between virus and host cell proteins are strong determinants of influenza A virus pathogenicity [29,30]. Once influenza virus gains access to the human respiratory tract, it attaches to and penetrates the epithelial cells via the HA protein and uses the host cell system for its replication. The newly emerging progeny viruses bud from cell membrane, are released with the assistance of NA, and infect adjacent epithelial cells. The NS1 protein inhibits IFN-I production in virus-infected cells by interfering with the RIG-I signaling pathway. Pulmonary macrophages then induce epithelial cell apoptosis, which is the underlying mechanism of viral pathogenesis and is mediated by the tumor necrosis factor (TNF)-related apoptosis-inducing ligand (TRAIL)-DR6 in the infected lung cells. After shedding, it causes local inflammatory responses as well as systemic toxic symptoms such as high fever, pain, and a decreased white blood cells (WBC) count. The infected cells can also produce excessive interferon (IFN), which may be related to the systemic symptoms. However, no viremia has been reported [31,32]. 

Oda et al., 1989, indicated that oxygen radicals produced by the host’s delayed response affect the mortality of virus-infected mice and are important for the pathogenesis of influenza virus infection [33]. The effect of influenza virus on the pro-/antioxidant balance in the host cells has been investigated and the virus has been shown to be able to generate reactive oxygen species (ROS) and pro-oxidant cytokines such as TNF from phagocytes [34]. Other studies did not support a direct role for ROS-induced tissue damage in the pathogenesis of influenza virus infection in the lungs of infected mice [35]. It was concluded that excessive production of NO, mediated by IFN-γ, together with O_2_-, which forms more reactive peroxynitrite, may be the most important pathogenic factors in influenza virus-induced pneumonia in mice [36]. High influenza viral load activates a burst in phagocytic cells. The infected mice lung cells exhibited oxidative stress via enhanced superoxide and xanthine oxidase (XO) generation, as well as decreased capacity of small molecular antioxidants. Cytokines produced in the lungs may contribute to the systemic effects of influenza. ROS may activate viral replication via activation of NFκB, while oxidants decrease the CD4+ T cell count by inducing apoptosis in these cells [37]. Furthermore, it has been suggested that the influenza virus activates phagocytes by the generation of metabolites derived from superoxide (O2) and nitric oxide (NO) after interaction between the viral surface glycoproteins and the phagocyte’s plasma membrane. The virus may take advantage of these potentially virucidal metabolites to promote infection. Murphy et al., 1998, showed that experimental influenza virus infection increased oral NO, but not nasal exhaled NO levels, suggesting that NO does not directly contribute to the respiratory tract symptoms after infection with influenza A virus [38].

With regards to surface glycoproteins, neuraminidase of the virus plays an essential role in the release of virus from the infected cells by cleaving the terminal sialic acid residue from the host receptor and helps to release the newly synthesized progeny viruses. This protein has a highly conserved active site in both type A and type B of influenza viruses and is the target of neuraminidase inhibitors. Resistance to neuraminidase inhibitors, which is accompanied by mutations in hemagglutinin, could affect the antigenicity of influenza viruses [39].

Influenza infection has been shown to increase ROS production and decrease antioxidant levels in the host by dysregulation of T helper (Th)1/Th2 cytokine balance and an increase in pro-inflammatory cytokines production. After influenza infection, IFN-α/β, IFN-α, and IL-2 appear to have protective roles against influenza infection, while IL-1, TNF-α, and IL-6 seem to be involved in the inflammatory phase of the infection [40]. IFN-α/β induces an antiviral state in cells by stimulating the transcription of genes coding for antiviral proteins such as 20,50-oligoadenylate synthetase (2–5 OAS), double-stranded RNA-dependent protein kinase, and the Mx proteins [41]. The protein 20,50-oligoadenylate, produced by the activity of 2-5 OAS, activates cellular endoribonuclease, which is responsible for single-stranded viral and cellular RNA degradation [41]. Protein kinase P1/eIF2 blocks viral protein synthesis and limits viral spread [42]. Human MxA protein inhibits influenza viral replication at post-transcriptional and translational levels [43]. IFNα/β also stimulates memory cells [44]. IFN-β, which is produced by Th1 and NK cells, stimulates macrophages and NK cell activity, upregulates the expression of major histocompatibility complex (MHC) molecules, and controls immunoglobulin class switching [45]. Meanwhile, IL-2, also produced by Th1, stimulates a rapid proliferation of T cells, which arrange many immunologic events to induce antibody responses [46]. The IL-1 family includes of IL-1α, IL-1β, and IL-1 receptor antagonist (Ra). IL-1α and IL-1β can induce fever, anorexia, hypotension, and sleepiness. IL-1Ra provides some protection against IL-1 by blocking the binding of IL-1 to its cell surface receptors [47]. Because of its cytotoxic and pro-inflammatory effects, TNF-α plays a dual role (beneficial and harmful to the patient) during influenza infection. It is produced by several immune cells, including monocytes, NK, T, B, and neutrophils [48]. In vitro TNF-α treatment leads to infectious virus decrease, viral protein synthesis inhibition, and cytopathic effect of virus reduction [49]. Systemic TNF-α administration produced fever and anorexia, which regulate body temperature and appetite. Other functions of TNF-α include induction of other cytokines (such as IL-1 and IL-6), induction of apoptosis in mature T cells and increased phagocytic activity [48]. IL-6, produced by lymphoid and nonlymphoid cells, is a multifunctional cytokine that regulates immune and acute phase responses and hemopoiesis through its involvement in T cell activation, growth and differentiation, as well as B cell differentiation [50].

In a case study by Jacoby et al., 2002, it was discussed that influenza virus infection may cause an increase in airway inflammation and bronchoconstriction, particularly during asthma attacks. Inflammatory mediators, which are produced by epithelial cells and increase in response to influenza virus infection, include IL-6 and IL-8, regulated on activation, normal T lymphocytes expressed (RANTES), eotaxin, oxygen radicals, and antioxidant enzymes [51].

It has been suggested that the cell cycle would be arrested under stress conditions by malondialdehyde (MDA), the endogenous product of lipid peroxidation. MDA can trigger p53 and p21 induction and may induce G1, S, and G2/M cell cycle arrest irreversibly [52]. Exposure of the vascular smooth muscle cell (VSMC) to HNE (the aldehyde component of oxidized low-density lipoprotein) causes mitogenesis by ERK, JNK, and p38 MAP kinase activation, induction of c-fos and c-jun gene expression, as well as activator protein-1 (AP-1) activity. ERK is involved in cellular proliferation and differentiation, whereas JNK and p38 MAP kinases perform powerful responses to cellular stress such as pro-inflammatory cytokines. Activation of JNK induces apoptosis in response to stimulating cell proliferation and transformation, while c-fos and c-jun activate the transcription of several genes controlling cellular growth. AP-1, which is encoded by c-fos and c-jun, is also believed to regulate genes involved in the control of cell growth and differentiation. Indeed, HNE induces VSMC growth, which is consistent with other observations. These observations are consistent with the role of lipid peroxidation products in vascular smooth muscle cell growth [53].

Palamara et al., 2005, also showed that influenza A virus infection causes the activation of various MAPK pathways, such as the p38MAPK and JNK pathways, which play roles in the inflammatory and apoptotic responses, and the Raf/MEK/ERK cascade, which results in nuclear transport of vRNPs. Phosphorylation events also play critical roles in virus penetration into the cell, efficient nuclear export of NP, and progeny viruses budding [54]. Influenza viruses use several strategies to recruit host cell machinery for their replication and infection. Among these, the imbalance of intracellular redox state plays an important role in modifying the activity of different signaling pathways. In particular, mild oxidative imbalance [55], ligand/receptor binding [56] and cytokines [57], cause localized oxidation of reactive cysteine residues of “redox sensitive” proteins, which reversibly activate or deactivate protein functionality [58].

It has been reported that influenza virus is propagated in the epithelial cells lining the respiratory tract. The virus-infected cells are lysed by the cytocidal action of CTLs and NK cells in the local mucosal membrane. One of the mechanisms by which CTL exerts cytolytic activity is via perforin, a pore-forming protein, which is a main constituent of cytotoxic proteins involved in the granule exocytosis pathway. It plays a crucial role in the host defense mechanism against influenza virus infection, especially in its early stage, as the local respiratory defense by inducing apoptosis of virus-infected cells [59].

The effect of different models of oxidative stress, such as immobilization, cold and cold-restraint stress, was investigated on liver monooxygenase activity and lipid peroxidation in influenza virus-infected mice. These models, in combination with influenza virus infection, were associated with graduated oxidative disturbances in the livers of mice, which might be related to the production of hepatotoxic mediators outside of liver and their transportation and diffusion to the liver. In addition, these models were accompanied with a significant increase in lipid peroxidation products, such as thiobarbituric acid reactive substances (TBARS), a decrease in natural antioxidants (including vitamin E, glutathione) and cytochrome P-450, a cytochrome c reductase, and liver monooxygenases [60]. At a later stage, following influenza A virus (A/Aichi/2/68/H3N2) infection, several disorganizations were recorded in lung and blood: Increase in total MDA in lung, progressive damage of the alveolar cells, acute inflammatory reactions, development of bronchitis and pneumonia, lung disorder associated with release of cytokines and massive infiltration of polymorphonuclear leukocytes into the alveolar space, conversion of xanthine dehydrogenase to xanthine oxidase, decrease in PaO_2_ and development of hypoxia, increase in PaCO_2_ and development of metabolic acidosis, eicosanoids and prostaglandin E2 and enhanced immune response, oxidative stress in lung tissue and blood, accumulation of methemoglobin in blood and nitric oxide in lung, free radicals production, increase in conjugated dienes, decrease in blood vitamin E, as well as intensive generation of free radicals mostly in the early stage of influenza virus infection in blood and lung. The promising effect of vitamin E supplementation was dose dependent in blood, but dose independent in lung [61]. 

Valyi-Nagy et al., 2012, highlighted that oxidative damage is a mechanism activated by viral infection, which may result from direct effects of virus on cells or indirect effects of host inflammatory responses. Inflammatory responses of the host cells augmented by viral infection may cause for O_2_- and other ROS production by infiltrating phagocytic cells and XO-mediated humoral responses. Inflammatory cytokines such as IFN-β lead to the induction of inducible nitric oxide synthase (iNOS) during influenza virus infection. ROS, reactive nitrogen species (RNS), and secondary lipid peroxidation products like HNE and MDA may affect viral replication through modulation of the activation state of cells, regulation of host immune responses, and oxidative damage to host tissues and viral components [62].

ROS production is an important host defense mechanism against influenza virus infection. However, excessive superoxide production may stimulate the release of pro-inflammatory cytokines and chemokines, such as TNF-α IL-6, IFN-γ, IL-8, monocyte chemoattractant protein-1 (MCP-1), macrophage inflammatory protein 2 (MIP-2), and MIP-1a, as well as inflammatory cells such as macrophages and neutrophils into the lung. This induces activation of the innate immune system, which is essential for viral clearance and resolution of inflammation, but might be harmful to the cell. Persistent trafficking of these inflammatory factors causes a cytokine storm, which may have a reverse effect on viral clearance and causes excessive damage. Suppression of superoxide production by targeting the enzymatic major source of superoxide and ROS in mammalian inflammatory cells, NADPH oxidase 2 (Nox2), reduces influenza A virus-induced lung injury and virus replication [63]. Superoxide anion, produced by macrophages infiltrated into the virus-infected organs, is used for the development of severe influenza-associated complications. Influenza-induced oxidative stress increases vascular permeability and breakability resulting in the airway mucosa and lung tissue edema, as well as multiple hemorrhages in the alveolar area, interstitial lung cells, and all internal organs. Therefore, the combination of antioxidants with antiviral drugs could synergistically reduce the lethal effects of influenza virus infections [64].

### 1.3. Medicinal Plants

Medicinal plants have been used extensively and throughout the world, even in regions with well-developed healthcare systems [65], and numerous traditional cultures still rely on indigenous medicinal plants for their primary health care needs [66]. Medicinal plants are increasingly regaining popularity in modern society as natural alternatives to synthetic medicines and they offer great potential as potentially effective new antiviral agents [67]. Traditional herbs are generally cheaper, more accessible, or readily available, and sometimes more culturally acceptable. Furthermore, some adverse effects of synthetic drugs [68,69] have made researchers focus on natural herbal substances as complementary therapies and preventive medicine. 

Natural active agents derived from medicinal plants often represent the most critical raw materials of lead compounds in pharmaceuticals. Although traditional societies have exploited medicinal plants to combat certain diseases, a crude extract dosage is not safe to prescribe, for two reasons: first, some phytochemicals may exist at toxic levels in crude extracts [70], and secondly, the bioactivity may be suboptimal if maximal activity requires specific combinations of phytochemicals [71]. Therefore, the WHO has recommended DNA barcoding for identification of medicinal plants and to facilitate the detection and quantification of the required chemical claimed compounds [72].

Medicinal plants are now universally recognized as a crucial component of human health care, as well as social and economic support systems [73]. Studies that have investigated the chemical profile and composition of medicinal plants reveal the complexity and variety of compounds, contributing to the various uses of plants in treating numerous diseases, including life-threatening diseases such as viral diseases and cancer [74].

### 1.4. Quercetin and Its Derivatives 

Quercetin (3,3′,4′,5,7-pentahydroxyflavone) is a natural compound from the flavonoid family of the glycoside rutin [75], which all share a similar flavone backbone (3-ringed molecule with OH groups attached). Several other substitutions have generated subclasses of flavonoids with different compounds within these subclasses. Quercetin has been isolated from more than 20 plant materials from the United States, European and Asian countries, and South Africa [76,77,78]. It is well known for its range of therapeutic properties, which include anti-inflammatory [78,79], antiproliferative [80], antioxidative [81], antibacterial [82], anticancer [83], neuroprotective [84], hepatoprotective [85], and antiviral [77,86,87] activities.

Quercetin, the most abundant flavonoid molecule [75], is itself the aglycone form of rutin, but flavonoids are bound to sugars in their natural state, and quercetin is thus also found naturally in its glycoside forms. Glycosylation can occur at any hydroxyl group of the compound [88]. The compound possesses 2 benzene rings (ring A and B), which are linked by a 3-carbon chain that forms a closed pyran ring (C ring) with variously distributed hydroxyl and methyl groups [89]. Table 1, adapted from Gansukh et al. (2014), with some modifications, provides a summary of the structural properties of quercetin and its derivatives [75]. It is worth noting that Pawlikowska-Pawlęga et al. [90], in their study, highlighted modifications in the phospholipid membranes of human skin fibroblasts (HSF) after exposure to quercetin, which was found to localize in the polar head regions of the membrane. It could be detected on membranes in the cytoplasm, around the nuclear envelope and inside the nucleus and nuclei by increasing the exposure time. This property affects its quality of pharmacological activities. Figure 2 shows the schematic mode of attack of quercetin and its derivatives on the influenza virus life cycle in different parts of the cell.

## 2. Materials and Methods

### 2.1. In Vitro and In Silico Studies against Influenza Virus

#### 2.1.1. General Overview

Recent studies have described the therapeutic effect of quercetin and derivatives acting against influenza virus. Choi et al., 2009, showed that quercetin 3-rhamnoside (Q3R) isolated from Houttuynia cordata has antiviral activity against influenza A/WS/33 virus in vitro [86]. According to ethnopharmacological studies, different parts of the *Schinopsis brasiliensis* plant, typically found in the northeastern Brazil, are widely used for the treatment of health disorders including influenza, diarrhea, and inflammation [91,92]. Ultrahigh-performance liquid chromatography coupled to high-resolution mass spectrometry (UPLC-QTOF-MSE) allowed the identification of 33 compounds in *S. brasiliensis*, including corilagin, chlorogenic acid, and quercetin derivatives, which may contribute to its antiviral activity [93]. The antiviral activity of *Tetrastigma hemsleyanum* Diels et Gilg, a widely used medicinal plant in China, was characterized against the recombinant influenza virus PR8-NS1-Gluc in MDCK cells with Gaussia luciferase viral titer assay. Quercetin and isoquercetin were clarified among the constituents using UPLC coupled with hybrid quadrupole-orbitrap mass spectrometry (UPLC-Q-Exactive/MS) [94].

In a study by Kim et al., 2010, selected polyphenols, including quercetin, isoquercetin, quercetin 3-glucoside, resveratrol, rutin (quercetin-3-rutinoside), quercetin-3-galactoside, quercetin 3-rhamnoside, kaempferol, luteolin, fisetin, catechin, epigallocatechin (EGC), and epigallocatechin gallate (EGCG) were tested for their antiviral activity against influenza A and B H1N1 viruses (A/swine/OH/511445/2007 (H1N1) and human A/PR/8/34 (H1N1), as well as human influenza B virus (B/Lee/40). The *Hypericum perforatum* and *Equisetum arvense* L. extracts were tested in parallel and analyzed by high-performance liquid chromatography (HPLC), thereby confirming the presence of quercetin and isoquercetin. In accordance with previously described findings, isoquercetin inhibited the replication of both influenza A and B viruses at the lowest 50% effective concentration (EC50). Isoquercetin, in combination with amantadine, exhibited synergistic effects on viral replication reduction in vitro, as well as suppressing the emergence of the resistant virus [95]. *Capparis sinaica* Veill methanol extract was tested for antiviral activity against H5N1 using plaque inhibition assay in MDCK cells. The polyphenol-rich extract showed 100% inhibition of viral activity at 1 µg/mL concentration. Using bioactivity-guided fractionation, the fractions eluted with ethyl acetate and 25% methanol in ethyl acetate were found to possess antiviral activity. Chromatographic separation of these fractions confirmed the isolation of quercetin, isoquercetin, and rutin from this species, which showed a reduction in the virus titer by 68.13%, 79.66%, and 73.22% at concentration of 1 ng/mL, respectively [96].

Yang et al., 2014, investigated the in vitro antiviral activity of tea polyphenol compounds against influenza A/PR/8/34 and B/Lee/1940 viruses. Tea polyphenols mainly contain catechins, proanthocyanidins, and theaflavins as active compounds, as well as flavonols such as kaempferol, quercetin, and myricetin, which are similar to catechins but have more planar chemical structure. The CPE inhibition assay in MDCK cells was used to evaluate the antiviral effect of the compounds in vitro. Quercetin with 42.28 µg/mL CC_50_, showed IC_50_ value of more than 42.28 in both influenza A and B virus treatments. This study did not show a strong anti-influenza effect of quercetin in MDCK cells as infection model compared to the other polyphenols, which may be due to different study designs and starting points [97]. 

Abdal Dayem et al. [89] showed anti-influenza activity of some flavonoid compounds such as diosmetin, eriodictyol, kaempferol, and isorhamnetin along with quercetin isolated from polyphenol extracts. Among these, isorhamnetin showed the most potent antiviral activity, but, quercetin, which was tested in different periods, also caused suppression of influenza virus-induced cell death. They evaluated the effects of isorhamnetin and quercetin both in 50 µM concentration in pre- and cotreatments. In a virus yield reduction assay, quercetin led to a significant decrease in virus titer [89].

Several active components were identified in *Vaccinium oldhamii*, belonging to the blueberry family, which inhibit influenza infection. Three fractions of 30%, 40%, and 50% ethanol were separated, which represented polyphenols identified by HPLC analysis. BY means of LC/MS analysis, procyanidin B2 and ferulic acid derivatives, ferulic acid *O*-hexosides, quercetin-3-*O*-rhamnoside, and quercetin-*O*-pentoside-*O*-rhamnoside compounds were identified. The inhibitory activity against influenza virus (A/Yamagata/165/2009/H1N1/pdm09) was calculated by a reduction in the number of plaques created by the virus. All the compounds which showed intense inhibitory activity in the early stage of flu infection were ferulic acid derivatives. However, quercetin glycosides showed weak activity in this study. Although anti-influenza activity of quercetin was reported previously [78,86], it was suggested in this study that this activity was due to the suppression of influenza replication inside the cells and not due to the inhibition of virus infection to the host cells [98].

To investigate the effect of structural changes on the quercetin activity, various C3, C3′, and C5 hydroxyl group substituted quercetin derivatives were synthesized with various phenolic ester, alkoxy, and aminoalkoxy moieties and their anti-H1N1 activities were investigated against A/swine/OH/511445/2007 (H1N1), Oh7. The synthesized C3-analog, quercetin-3-gallate, which was structurally related to EGCG, showed favorable antiviral activity against influenza virus in MDCK cell culture comparable to that of EGCG, reducing cytopathic effect (CPE) in the cells with higher in vitro therapeutic index. The results imply that further modification at C3 functional group may play an essential role in improving the compound efficacy [99]. 

The structure–activity relationship (SAR) of quercetin was analyzed in the Yang et al. [97] to investigate the antiviral activity of polyphenol-rich extracts containing quercetin against influenza A/PR/8/34 and B/Lee/1940 viruses. Two mechanisms were assumed for the antiviral activity of compounds on both influenza A and B viruses, i.e., one is at the early stage by blocking the viral binding to the cell receptors, whereas the other is after the entry by attenuation of viral replication. It has been shown for in vitro NA inhibition assay that rigid flavonols (kaempferol, quercetin, and myricetin) have stronger inhibitory effects than flexible ones (catechin and epicatechin) [100]. One hydroxyl at C-4′ position in the B ring, a double bond at C-2–C-3, and a ketone at the C-4 position is essential for inhibiting influenza virus as NA inhibitor. However, the presence of more hydroxyl in the B ring, such as quercetin and myricetin, causes steric interference and disfavors the anti-influenza effect [100]. It seems that dimeric compounds are more potent antivirals against both influenza A and B viruses than the monomers such as catechin and quercetin [97].

Regarding structural variations, the reaction of quercetin with bromine was also evaluated under various conditions like glacial acetic acid at 35–40 °C and absolute ethanol at 0–5 °C and 20–22 °C, which yielded with more than 90% of purity compounds. The compound QR 6,8-dibromide (52–54% yields, 96–98% purity by HPLC) exhibited moderate inhibitory activity against pandemic influenza virus A/H1N1/pdm09 in MDCK cells (EC50 6.0 µg/mL, CC_50_ 97.7 µg/mL, selectivity index: 16) [101].

#### 2.1.2. No Activity against Influenza Virus

Acetone extract of the shoots of *Helichrysum melanacme* (a traditional medicinal plant used in South Africa to combat cough, fever, headache, colds, and chest pain) was subjected to in vitro bioassay-guided fractionation using human influenza virus A/Panama/1/68 (H3N2). Several compounds were isolated, including 3-*O*-methylquercetin and quercetin, although no 50% inhibitory concentration (IC_50_) value could be obtained. These compounds showed no activity against influenza virus as evaluated by direct immunofluorescence [102]. 

*Rumex pictus* is an edible plant used for medicinal purposes around the world [103]. Quercetin-3-*O*-β-*D*-glucouronide, along with some other flavonoids (rumpictuside A, apigenin 7-*O*-β-*D*-glucoside, vitexin, orientin, and isorientin), isolated from this plant were characterized by UV, FT-IR, 1D, 2D NMR, and HR-FAB-ESI-MS. Although most of these compounds showed significant antioxidant activity against DPPH and ABTS, and having been previously reported to show evidence of antiviral activity, they were found to be inactive against influenza A virus infection in this study [104]. 

The anti-influenza activity of preparations of compounds with antioxidant and/or antihypoxant properties, such as hypoxene, reduced glutathione, dihydroquercetin, Trolox, coenzyme Q10, and the enzymatic preparation of superoxide dismutase, alone or in combination with rimantadine, was investigated with respect to influenza A (H3N2) infection in cultures of chorioallantoic membranes of chicken embryos and in MDCK cells. The antiviral activity of the aforementioned preparations was assessed by monitoring inhibition of the virus cytopathogenic effects on the cells. Among these preparations, free radical scavengers, including coenzyme Q10, Trolox, quercetin, and the enzymatic preparation superoxide dismutase, did not show protective effect and in some cases, even enhanced the production of viral particles and decreased the antiviral action of rimantadine [105].

#### 2.1.3. Anti-Inflammatory and Immunomodulatory Effects

Shikimic acid, one of the primary components of oseltamivir, alone and in combination with quercetin were evaluated in comparison with oseltamivir in an in vitro model in terms of their ability to modulate the expression of IL-6 and IL-8 as immunomodulatory cytokines from peripheral blood mononuclear cells (PBMCs) [106]. IL-8 attracts T-cells to the site of infection [107] and applies indirect antiviral activity due to the recruitment of leukocytes [108]. IL-6 is secreted at the early stages of the inflammatory process by activated monocytes and has been reported to act as a positive immunomodulator by empowering innate immunity [109]. Peripheral blood mononuclear cells (PBMCs) isolated from six healthy volunteers were treated with different doses of shikimic acid with or without quercetin. The IL-8 and IL-6 expression levels were measured by means of ELISA. It was shown that shikimic acid alone was not able to modulate innate immunity in terms of antiviral activity. However, its combination with quercetin, even at low doses, effectively modulated innate immunity by increasing the IL-8 and IL-6 levels [106]. 

The potential radical scavenging capacity, as well as anti-inflammatory and anti-NA properties of 13 compounds, isolated from Chaenomeles speciosa, was evaluated. *Chaenomeles speciosa* (Sweet) Nakai is a traditional herb in China. Its dried fruit is widely used for the treatment of rheumatoid arthritis, prosopalgia, and hepatitis, as well as being used as an appetizer. Studies have confirmed the presence of glycosides, flavones, phenolics, tannins, and organic acids in this fruit, including quercetin. Quercetin showed significant dose-dependent DPPH radical scavenging activity, with an IC_50_ value of 3.82 µg/mL. Along with other compounds, it also inhibited the NO production by more than 25% at a 5 µg/mL dose. It could inhibit the production of TNF-α by 33.14% compared with the control (*p* < 0.05). It also had a modifying effect on the release of IL-6 in RAW264.7 macrophage cells, with an inhibitory rate of 39.79% (*p* < 0.05). Furthermore, quercetin was spectrofluorometrically shown to exhibit activity against NA of A/PR/8/34 (H1N1)) with an IC50 value of 1.90 µg/mL. As a powerful antioxidant, quercetin from this fruit was suggested as a potential antiviral and anti-inflammatory agent [110].

The aqueous extract of *Epimedium koreanum* Nakai, a traditional Korean and Chinese plant that is used for medicinal purposes, was evaluated against different viruses, including influenza A virus (PR/8/34), in RAW264.7 and HEK293T cells). The primary component of this extract was found to be quercetin, as determined by HPLC. Application of the extract induced the secretion of type I IFN and pro-inflammatory cytokines, stimulating the antiviral activity in the cells. It was suggested that this extract plays a role as an immunomodulator of the innate immune response and may be recommended as prophylactic or therapeutic treatment against different viruses [111].

The inhibitory effect of Epimedium koreanum Nakai toward influenza A/PR/8/34 was shown by reduction in cell death and viral titer by nearly 1.8-fold at 24 h postinfection (hpi) in RAW264.7 cells. The CC_50_ of the extract in these cells was 14.6 ± 1.68 µg/mL and EC50, determined by infection with PR8-green fluorescent protein (GFP) (MOI =1.0) and measuring the GFP expression at 24 hpi with the GloMax-Multi Detection System, was 0.94 ± 0.23 µg/mL. The high SI value (15.5) supported promising prophylactic or therapeutic potential of the extract, which stimulated the immune cells and induced high levels of cytokines such as IFN-β and IL-6, which mediate the antiviral state in the cells. In addition, treatment of cells with this extract upregulated the phosphorylation of key signaling molecules in the type I IFNs and NF-κB pathways; IRF-3, STAT1, TBK1, p65, as well as p38 and ERK, which are involved in controlling viral infection. The phosphorylation of these molecules was comparable to that obtained with LPS treatment, a potent stimulator of TLR4. The transcriptional levels of various antiviral genes, including Mx1, IFN-β, P56, IRFs, GBP-1, PML, IL-6, OAS-16, ISG-15, and ISG-56, were also upregulated by treatment with this extract [111]. 

Choi et al. [112] observed induction of immunomodulatory signaling molecules (IL-6, IFN-β, and TNF-α and antiviral activity of aqueous extract of *Eupatorium fortunei* against influenza A virus, Newcastle disease virus, and vesicular stomatitis virus in murine RAW264.7 macrophage cells [112]. Fifteen active compounds, including quercetin, psoralen, and quercitrin, were detected by UPLC-MS/MS. Pretreatment with *Eupatorium fortunei* aqueous extract markedly reduced viral replication, as evaluated by the GFP-tagged virus. It also induced the production of type I IFN, including pro-inflammatory cytokines. This extract regulated the phosphorylation of IRF-3, STAT1, and TBK1, which are key molecules in the signaling pathway of type I IFN. This extract and its active components were introduced as immunomodulators of the innate immune response in murine macrophages as a basis for the development of prophylactic or treatments against viruses. Influenza replication was dramatically suppressed in quercetin-treated cells [112]. 

*Rapanea melanophloeos*, a South African medicinal plant of the family Myrsinaceae, was investigated for the first time in as a source of quercetin-3-*O*-α-*L*-rhamnopyranoside and for its immunomodulatory properties against influenza A virus. This plant has traditionally been used to treat respiratory ailments [78]. A previous preliminary report described anti-influenza activity exhibited by a methanol extract of this plant, protecting cell viability and viral titer decrement [113]. Quercetin-3-*O*-α-*L*-rhamnopyranoside was purified from fraction FIII on Sephadex LH-20 column chromatography using methanol. Simultaneous, prepenetration, and postpenetration combination treatments of quercetin-3-O-α-L-rhamnopyranoside (at a nontoxic concentration of 150 µg/mL) and A/PR/8/34 H1N1 virus were evaluated over 1-h incubation in MDCK cells. Using HA and qPCR assays, the virus titer and viral load targeting NP and M2 viral genes were determined, respectively. TNF-α and IL-27 levels were also determined, as major pro- and anti-inflammatory cytokines. In summary, quercetin-3-*O*-α-*L*-rhamnopyranoside was found to decrease the viral titer by 6 log fold, reduced viral genome expression (*p* < 0.01) and affected cytokine levels significantly [78]. 

Further research focused on the specific mechanism of virus–host interactions. MDCK cells were exposed to various combinations of quercetin-3-*O*-α-*L*-rhamnopyranoside and H1N1. Other cytokines (IL-6 and CCL-2, both pro-inflammatory cytokines, and IFN-β as anti-inflammatory cytokine) were evaluated at RNA and protein levels. The expression of IL-6 and CCL-2 at genome and protein levels were significantly affected by quercetin-3-*O*-α-*L*-rhamnopyranoside, especially upon copenetration treatment. Regarding IFN-β, no significant difference was observed at the protein level between H1N1 inoculation and combined treatments. It can be due to intracellular viral NS1 protein that prevents the induction of beta interferon [114]. In addition, an increase in TNF-α inhibits the production of IFN-α/β. These might be the reasons why quercetin-3-*O*-α-*L*-rhamnopyranoside did not act through IFNs [78]. 

#### 2.1.4. Influence on Molecular Pathways

Influenza virus recruits a number of host cell components, activating different signaling pathways, for its passage into the cell, as well as for propagation and replication [115]. Several studies have investigated the potential of quercetin and its derivatives to counteract signaling pathways used for influenza virus propagation.

Quercetin and oseltamivir were evaluated for their effect on the Toll-like receptor 7 (TLR7) signaling pathway in dendritic cells and macrophages infected with H1N1. Leukomonocytes were obtained from umbilical cord blood and harvested after stimulation using recombinant human granulocyte-macrophage colony-stimulating factor (rhGM-CSF) and the recombinant human IL-4. Human bronchial epithelial cells (16HBE) were infected with H1N1 and cocultured with the immunological cells, following which quercetin and oseltamivir were added into the medium. The expression levels of genes related to TLR7 pathway were determined by PCR. Quercetin and oseltamivir increased cell viability and inhibited the TLR7 signaling pathway induced by viral infection [116].

Wan et al., 2013, investigated the effect of quercetin on cyclin-dependent kinase 4 (CDK4) at mRNA and protein level in A549 human lung epithelial cells infected by influenza A H1N1 (A1/Qian fang/166/85) in parallel with ribavirin. The CDKs are involved in cell division and any dysregulation in their expression by infection causes DNA damage and apoptotic process. The cells were infected with virus for 2 h, following which the maintenance media containing quercetin (10 µg/mL) and/or ribavirin (20 µg/mL) was added. The cells were then harvested after 4, 12, 24, and 48 h of culture, and total RNA and protein were extracted. The CDK4 mRNA and protein expression, measured at four time points, increased significantly in the H1N1 group (*p* < 0.01), but decreased significantly in response to treatment with quercetin and ribavirin (*p* < 0.01). It seems that quercetin can induce DNA repair mechanisms in A549 cells infected with H1N1, contributing to host cell proliferation. Quercetin was not as potent as ribavirin in promoting cell survival, but it was less toxic and inhibited the mRNA and protein expression of CDK4 induced by H1N1 infection. It was suggested that quercetin possesses potential clinical antiviral application value [117]. 

Autophagy is one of the pathways that is essential for the host cell to remove unnecessary or dysfunctional components by degrading and recycling cellular components [118]. Autophagy is a key host cell response to viral infections, including influenza virus infection [115]. However, it has been shown that influenza A virus is able to inhibit autophagy [119], which causes the accumulation of autophagosomes by blocking autophagosome maturation and union with lysosomes using ion channel M2 protein, which is necessary for fusion [119,120]. Therefore, managing this process can inhibit IAV proliferation and may be a strategy for developing new anti-IAV agents.

Influenza virus polymerase consists of PA, PB1, and PB2 subunits and plays an essential role in viral RNA synthesis [121]. It has been suggested as a target for influenza drugs, as it is highly structurally conserved across different influenza virus types and strains [122]. The *Dianthus superbus* is a medicinal plant used in Mongolia and China for its anti-inflammatory properties. The quercetin-7-*O*-glucoside, isolated by structural elucidation by means of 1H-NMR and 13C-NMR, was evaluated for efficacy against virus-induced symptoms, ROS production, and autophagy induction. MDCK cell line was infected with influenza viruses A and B (A/Vic/3/75 (H3N2), A/PR/8/34 (H1N1), B/Maryland/1/59, B/Lee/40) and subjected to time-dependent assays, including pre- and postincubation with quercetin-7-*O*-glucoside. The IC_50_ values obtained were 6.61, 3.1, 5.17, and 8.19 µg/mL, respectively. Quercetin-7-*O*-glucoside reduced ROS and autophagy formation by affecting cellular pathways but not the virus particles directly. Molecular docking analysis showed an inhibitory effect of quercetin-7-*O*-glucoside on viral RNA polymerase by occupying the binding site of m7GTP on viral PB2 protein [81].

Vaidya et al., 2016, evaluated the antiviral potential of biochemical components present in Kimchi, with particular focus on quercetin. Kimchi is a traditional fermented food commonly consumed in Korea. Its primary ingredients are onion, garlic, ginger, salt, and red pepper. It contains various antimicrobial compounds such as capsaicin, allicin, and quercetin [123]. The quercetin content of Kimchi increases during fermentation. The authors investigated the effect of periodic quercetin treatments against influenza A virus H1N1 (low pathogenic human IAV H1N1 strain A/PR/8/34) in MDCK cells by evaluating the protein expression modulation. Significantly higher cell viability, lower expression of the influenza PA gene, reduced apoptosis, and reduced cell death were observed in cells subjected to periodic treatment compared with cells at the pretreatment stage. A comparative proteomics analysis approach was used to examine 220 proteins so as to elucidate the complex interactions between the virus and the host. The levels of 56 proteins were differentially modulated by quercetin treatment in influenza-infected cells, and post-translational modifications were identified in a total of 68 proteins. The periodic treatment of quercetin was influential in the regulation of key protein expression. The major regulatory proteins identified in this study included FN1, HSPs, PHB, TUBA1C, and TUBA4A, all of which are associated with cell and IAV interaction network. Moreover, post-translational modification of CCT4, EIF3A, HSPA2, and TUBB2B could also be involved in controlling IAV replication [124].

The frequency of apoptosis was examined in response to simultaneous combination treatment with quercetin-3-*O*-α-*L*-rhamnopyranoside and influenza A/PR/8/34 by detecting RhoA GTPase protein and caspase-3 activity. This combination increased caspase-3 activity while decreased RhoA activation [87]. 

Choi et al. [125] investigated the antiviral activity of Aloe vera ethanolic extract and its components quercetin, catechin hydrate, and kaempferol, all identified by UPLC-MS/MS, against H1N1 and H3N2 IAV strains in vitro [125]. A variety of pharmacological properties have been reported for Aloe vera, including antibacterial [126] and anti-IAV properties [127]. The aqueous extract of *Aloe arborescens* significantly reduced the proliferation of IAV and influenza B virus (IBV) in vitro [127]. The antiviral activity of aloe-emodin against H1N1 strain was related to the restoration of NS1-inhibited STAT1-mediated antiviral responses in the transfected cells [128]. Choi et al. [125] showed in their study that this extract inhibited autophagy, which was otherwise induced by influenza A virus in MDCK cells, while treatment with individual extract components (including quercetin) inhibited M2 mRNA and protein expression. 

In a recent study, in vitro biochemical and toxicological evaluation was performed to evaluate the effect of quercetin 3-glucoside, isolated from *Dianthus superbus* L and verified by spectroscopic analysis, against influenza A and B virus infection. This plant is a well-known traditional herbal medicine used by ancient Mongolian and Chinese for its anti-inflammatory properties [129]. It exhibited potent antiviral activity, delivering IC_50_s at 4.93, 6.43, 9.94, 8.3, and 7.1 µg/mL toward A/PR/8/34, A/Victoria/3/75, A/WS/33, B/Maryland/1/59, and B/Lee/40, respectively. Quercetin 3-glucoside blocked influenza A/PR/8/34 replication in a time-dependent assay (−2 and −1 h before infection; at the time of infection; and 1, 2, 4, 7, 10, 15, and 24 h after infection) and decreased expression of the related genes M, Atg-5, and LC3-β. Virus-induced ROS production was reduced by quercetin 3-glucoside. It significantly decreased autophagy, detected by Acridine Orange (AO) staining, by blocking virus infection-induced acidic vesicular organelles (AVO) [130]. 

#### 2.1.5. Neuraminidase Inhibitory Activity

The inhibitory potency of several flavonoid derivatives has been evaluated on neuraminidase activity (NA) of different types of influenza virus. In a study by Rakers et al., 2014, the neuraminidase inhibitory activity of flavonoids was investigated against NA of A/California/7/09 (A/H1N1/pdm09), A/Perth/16/09 (A(H3N2))) and B/Brisbane/60/08 compared to the oseltamivir. The experimental findings and molecular dynamics simulations confirmed moderate inhibition of the flavonoids on NA activity. The extent and sites of glycosylation of the flavonoids showed no significant influence on their inhibitory potency [131]. 

To verify the moderate antiviral activity of quercetin, the antiviral efficacy of flavonol glycosides isolated from 95% aqueous ethanol extract of the leaves of *Zanthoxylum piperitum* was investigated against influenza A/NWS/33 (H1N1) virus in vitro. Several properties such as anti-inflammation, antioxidation, vasorelaxation, antibacterial, and antiviral have been proposed for this plant. The compounds quercetin 3-*O*-β-*D*-galactopyranoside, quercetin 3-*O*-α-*L*-rhamnopyranoside, and kaempferol 3-*O*-α-*L*-rhamnopyranoside were obtained by chromatographic separation, enriched in the ethyl acetate fraction as determine by biochemical activity. Plaque reduction and NA inhibitory assays were used to assess the antiviral activity, and NA activity decreased significantly in the presence of these compounds. Although their effect was not so strong as oseltamivir, they have the capacity as antiviral agents against the influenza A virus [132].

Six flavonoids (quercetin-3-sophoroside, kaempferol-3-sophoroside, kaempferol-3-sambubioside, kaempferol-3-neohesperidoside, kaempferol-3-glucoside, and luteolin) and one alkaloid (chelianthifoline) were isolated from bee pollen methanol extract of Korean *Papaver rhoeas* plant and characterized by nuclear magnetic resonance (NMR) and mass spectrometry [133]. Bee pollen is used in many regions of the world for its medical capacities against flu, ulcers, colds, and anemia [134]. It is rich in vitamins, lipids, carbohydrates, proteins, minerals, and various organic compounds. Some biological activities such as antioxidation, anti-inflammation, antifungal, anticancer, antiallergy, chemopreventive, and antimutagenic activities have also been reported. All these compounds showed NA inhibitory activity against H1N1, H3N2, and H5N1 and reduced the cytopathic effects of these viruses in MDCK cells. The inhibition of NA by quercetin-3-sophoroside had IC_50_ values at 88.3 ± 3.0, 75.1 ± 8.7, and 112.8 ± 8.2 µM for H1N1, H5N1, and H3N2, respectively. The selectivity index obtained was about 8. Lee et al. [133] suggested that a bulky sugar moiety on quercetin-3-sophoroside and other flavonoids decreased their NA inhibitory activity compared to the aglycone compounds.

*Dianthus superbus*, which is a traditional medicinal herb widely distributed and used in various Asian countries, was proposed as a candidate food with medicinal and therapeutic potential. The dried materials were extracted using 100% and 80% methanol. Chemical analysis was done on ethyl acetate, butanol, and distilled water extracts using LC-MS/MS. The butanol extract showed potent antiviral activity against both influenza A and B viruses (A/PR/8/34 and B/LEE/40) with IC_50_ values of 4.97 and 3.9 µg/mL, respectively. Both quercetin-3-rutinoside and isorhamnetin-3-glucoside showed high NA inhibitory activity [135]. 

#### 2.1.6. In Silico and Docking Studies

Quercetin, baicalein, chlorogenic acid, and oleanolic acid were tested in molecular docking to explore their binding ability to NA of H7N9. They showed high binding potential with NA, which even exceeds that of oseltamivir. The mutation R294K caused conformational changes in NA structure and induced oseltamivir resistance, while other compounds showed stable binding with mutated NA, which renders them potential agents to overcome the drug resistance of the mutant virus [136].

Quercetin and chlorogenic acid, potential lead compounds derived from traditional Chinese medicine, were tested in molecular docking against influenza A virus H1N1 (A/PR/8/34). They showed strong binding abilities (−10.23 and −11.05 kcal/mol for quercetin and chlorogenic acid, respectively) with NA of this virus, which were comparable with oseltamivir (−11.24 kcal/mol). To validate the NA inhibitory effect of these compounds in vitro, NA activity assay and cytopathic effect assay were carried out. The NA activity by quercetin and chlorogenic acid was almost fully inhibited under 50 µg/mL concentration and the viral cytopathic effect reduced significantly [137].

Recently, Naïve Bayesian, recursive partitioning and CDOCKER methods have been used to construct a compound-protein interaction-prediction virtual screening system of the virus–host interaction for new drug discovery against IAV. This system was used to predict active compounds in Yizhihao, one of the common Chinese herbal formulas used to treat influenza and cold symptoms. It consists of *Radix isatidis*, *Folium isatidis*, and *Artemisia rupestris*. The effectiveness of this formula has been shown against influenza A (H1N1, H3N2, oseltamivir resistant H1N1, and amantadine-resistant H3N2) and B strains in vitro, but with no report of ingredients and mechanisms [138]. The results from CPE reduction assay in this study provided scientific information for its usage and showed that quercetin, along with other compounds such as acacetin, indirubin, tryptanthrin, luteolin, emodin, and apigenin, had protective effects against wild-type strains A/PR/8/34 (H1N1) and A/Minfang/151/2000 (H3N2) with lower IC_50_ values and stronger activities than oseltamivir and ribavirin, especially when the viruses were preincubated and treated simultaneously with the drug compounds. Quercetin supports the activity of the compounds by reducing H1N1 viral activity or impairing viral adsorption [138]. NA glycoprotein has been considered an important target in drug design as it has a relatively preserved active site and is important in virus distribution. In a docking study by Sadati et al., 2019, the flavonoids quercetin, vitexin, chrysin, catechin, luteolin, naringenin, kaempferol, and hispidulin were selected to evaluate their potential inhibitory effect on influenza H1N1 virus by looking for their binding energy to the NA active site structure along with oseltamivir. These compounds showed high potential and affinity for binding to the active site of NA domain N1 with lower binding energies (−6.8 kcal/mol for quercetin, −7.2 kcal/mol for catechin, −6.9 kcal/mol for naringenin, −7.1 kcal/mol for luteolin, −6.8 kcal/mol for dinatin, −7.5 kcal/mol for vitexin, −6.8 kcal/mol for chrysin, and −6.8 kcal/mol for kaempferol) than oseltamivir (−5.8 kcal/mol). They were suggested to block the NA active site effectively [139]. In computational modeling, strong binding of quercetin-3-*O*-α-*L*-rhamnopyranoside with M2 transmembrane NA was observed for 2009 pandemic H1N1, N1, and H1 of PR/8/1934 and human RhoA proteins, with docking energies of −10.81, −10.47, −9.52, −9.24, and −8.78 kcal/mol, respectively, which makes quercetin-3-*O*-α-*L*-rhamnopyranoside a potent anti-influenza candidate [87]. Docking simulation using AutoDock Vina Software indicated a higher binding affinity of quercetin (−5.35 kcal/mol) as well as catechin hydrate (−5.48 kcal/mol) for the M2 protein than amantadine (−4.52 kcal/mol). Thus, the inhibition of autophagy induced by influenza virus infection may explain the antiviral activity of Aloe vera ethanolic extract against H1N1 or H3N2. The antiviral activity of Aloe vera ethanolic extract may go partially to catechin hydrate and quercetin [125]. Another molecular docking study revealed higher binding activities towards influenza polymerase and surface glycoproteins for flavonol glycosides [135]. Molecular docking studies highlighted the blockade of the cap-binding domain of polymerase basic protein subunit, showing higher binding affinity (−8.0 kcal/mal) than GTP (−7.0 kcal/mol) [130].

With respect to mutant viruses, A/HebeiXinhua/SWL1106/2017 (oseltamivir and amantadine-resistant H1N1) and mutant A/FujianXinluo/SWL2457/2014 (amantadine-resistant H1N1), quercetin, luteolin, and apigenin could more effectively reduce viral activity or impair the adsorption of viruses on the cells compared to amantadine and oseltamivir [140]. This study provided an experimental foundation for the development of broad-spectrum antiviral drugs. It also provided an efficient multitarget predictive tool for the discovery of new drugs against influenza [140].

#### 2.1.7. Hemagglutinin Inhibitory Activity

Quercetin has been reported to counteract influenza infection in the early stages by interacting with the HA2 subunit of influenza HA protein and inhibited virus-cell fusion. Quercetin exhibited an apparent inhibitory effect against both H1N1 and H3N2 virus strain infections in a dose-dependent manner. It showed IC_50_ values of 7.756 ± 1.097, 6.225 ± 0.467, and 2.738 ± 1.931 µg/mL against A/PR/8/34 (H1N1), A/FM-1/47/1 (H1N1), and A/Aichi/2/68 (H3N2), respectively. It also reduced the HA mRNA transcription and NP protein synthesis in a dose-dependent manner in MDCK cells. A time-of-addition assay was performed to identify the effective stage by measuring HA mRNA and protein levels at the stages of virus entry (0–2 h), replication and translation (2–8 h), and/or release (8–10 h). The best performance was observed in the entry and whole life cycle steps including attachment, endocytosis, and fusion. It was more effective during the process of virus infection rather than postvirus infection and was more effective on virus particles rather than the host cell. The binding affinity between HA protein and quercetin was confirmed by surface plasmon resonance (SPR) assay, which showed a medium-binding affinity of quercetin to HA protein, with a KD value of 1.14 × 10–3 µg/m. For further confirmation, the microscale thermophoresis (MST) assay was conducted, which showed that quercetin interacted with influenza HA protein in a dose-dependent manner. Using the pseudovirus-based drug screening system, quercetin also found to inhibit the entry of the H5N1 virus, which confirmed the specific interference with the function of the H5N1 influenza HA envelope protein [141].

### 2.2. In Vivo and Human Studies against Influenza Virus

The therapeutic effects of quercetin and derivatives have been evaluated in animal models as well as humans and have been studied from different aspects. One of the first in vivo studies on quercetin was conducted by Eşanu et al., 1981, who investigated the effect of aqueous extract of propolis, containing rutin and quercetin, on the experimental infection of mice with influenza virus A/PR8/34 (H1N1). When the extract was administered, 3 h after virus inoculation but not preinfection, the HA titers in the lung suspensions were reduced, and a slight decrease in mortality and an increase in survival length was observed [142]. The protective effect of quercetin was investigated during influenza virus-induced oxidative stress in adult male Swiss albino mice infected with influenza virus (A/Hong Kong/8/68). The mice were infected intranasally with 75 HA units of virus under anesthetic conditions and quercetin was administered orally at 1 mg/mL. The pathological effects were evaluated on day 8 postinfection. Upon assessment of lipid peroxidation and superoxide production, lipid peroxidation showed an increase in virus-infected mice, which decreased after administration of quercetin. Antioxidant enzymes superoxide dismutase (SOD) and catalase decreased in the lungs of virus-infected mice, detected by SOD enzyme assay and catalase assay, but increased in quercetin-treated mice at the same level as in the control group. It concludes that quercetin was effective in ameliorating the oxidative stress induced during influenza virus infection [143]. In addition, quercetin reduced the oxidative stress induced by influenza virus A/Udorn/317/72 (H3N2) infection in mice. Alveolar macrophages were obtained from the lung and superoxide radical production was assessed using nitro blue tetrazolium (NBT) reduction assay. The level of lipid peroxidation (LPO) was measured by TBARS and a significant decrease was observed in LPO levels after quercetin administration compared to untreated influenza infection [144]. Quercetin protects the lungs by maintaining the levels of endogenous antioxidant enzymes, including catalase, reduced glutathione, and SOD [145]. 

Savov et al. [146] examined the effects of rutin and quercetin on oxidative stress in healthy and the experimental influenza virus (A/Aichi/2/68 (H3N2))-infected mice. These flavonoids were injected intraperitoneally 3 days before and 2 days after the viral inoculation. Flu infection caused ROS and free radical generation in the 5th day after the viral inoculation. Viral infection increased the concentration of TBARS, suppressed cytochrome P-450, inhibited the activity of NADPH-cytochrome c reductase, slightly induced the activity of amidopyrine-N-demethylase (APND), and decreased analgin-N-demethylase (ANND) enzyme and monooxygenase activities. Although flavonoids are known for their antioxidant activities, rutin and quercetin exhibited pro-oxidant effect and caused oxidative damage in healthy mice, but still showed antioxidant activity in infected animals. Antioxidant effect of quercetin was stronger than rutin on the content of TBARS in the lungs, liver, brain, and blood plasma of influenza-infected mice. This biphasic effect might be due to several factors such as metabolic and physiological activities, as well as chemical structure. Thus, both were recommended for the treatment after infection, but not for prophylaxis and protection [146].

H5N1 virus induces pro-inflammatory cytokines, including TNF-α and IL-6 in monocytes and macrophages and in human alveolar and bronchial epithelial cells, to a greater extent than H1 and H3 subtypes [147,148]. Therapeutic or prophylactic strategies need to take these processes into account. A carefully designed nutritional supplement formulation including a combination of selenium, vitamin E, *N*-acetyl-cysteine/glutathione, resveratrol, and quercetin has been suggested based on the properties of each compound to counteract the pathogenic processes of H5N1 influenza in humans [149]. 

It has been shown previously that exercise stress can increase the risk of and susceptibility to upper respiratory tract infections [150,151], which are associated with a decline in macrophage antiviral resistance, which can be a problem for athletes and military personnel [152]. Short-term quercetin feeding was suggested as an effective strategy to reduce susceptibility to upper respiratory tract infection following stressful exercise in the controlled experimental study. The researchers subjected 4-week-old male ICR mice to running on a treadmill for 3 consecutive days to the point of fatigue (140 min). Quercetin (12.5 mg/kg) was given via gavage for 7 days before viral challenge. At 30 min, after the last set of exercise or rest, mice were inoculated intranasally with a standardized dose (0.04 hemagglutinating units) of influenza virus A/PR/8/34 (H1N1) and monitored daily for 21 days. The exercise was associated with increased susceptibility to infection, based on morbidity, mortality, and symptom severity on days 5–7 (*p* < 0.05), but quercetin treatment reduced this susceptibility to infection [153]. 

Intraperitoneal administration of isoquercetin to 4-week-old female BALB/c mice infected with human influenza A virus showed a significant decrease in virus titers and pathological changes in the lung. The lungs were subjected to RNA extraction and qRT-PCR to detect IFN-γ RANTES and iNOS levels as indicators of lung inflammation and virus titers. In addition, they were evaluated for histopathology [95].

The effect of quercetin-3-O-β-D-glucuronide isolated from Polygonum perfoliatum L., a well-known traditional Chinese medicinal plant, was investigated on influenza A virus in infected mice. A practical, specific, sensitive, reproducible, and accurate HPLC coupled with diode array detector (HPLC–DAD) method was developed to quantify quercetin-3-*O*-β-*D*-glucuronide in this plant. The compound was able to suppress lung edema induced by influenza A virus. In 3 and 6 mg/kg concentrations, it could inhibit edema caused by virus at 28% and 20%, respectively, compared with the control [154]. 

Confirming their previous in vitro results, Choi et al., 2012 evaluated the effect of quercetin 3-rhamnoside (Q3R) on influenza A/WS/33 virus-infected mice. The mice were orally treated with Q3R (6.25 mg/kg per dose) at 2 h before and once daily after influenza virus infection for 6 days. They exhibited a significant decrease in weight loss, necrosis, inflammation, and mortality, while their lung virus titers at 6 days after infection were about 2000 times lower than placebo-treated control mice and two times lower than oseltamivir-treated mice. The administration of Q3R delayed the development and progression of pulmonary lesions, which was confirmed with histological evaluation. They introduced Q3R as an attractive lead for the development of antiviral agents against influenza virus [155].

The protective effect of Epimedium koreanum Nakai against lethal doses of different types of highly pathogenic influenza A viruses (H1N1, H5N2, H7N3, and H9N2) was determined in BALB/c mice [111]. In the experimental groups, the mice were orally administered 0.1 mg/mL Epimedium koreanum Nakai at 1, 3, and 5 days before infection to confirm the prophylactic effects of the extract. Among the various components present in the extract, quercetin, in particular, was shown to have prominent antiviral properties. The mice were then intranasally infected with 5 times lethal dose (5MLD50) of each virus. The virus titers were determined by 50% tissue culture infectious dose (TCID50) from the lung tissues. All treated groups showed reduced viral titers. The extract-treated mice showed much lower body weight loss compared with the untreated group between 5 and 7 days postinfection (dpi) and recovered their weight loss by 8 dpi, and then returned to their normal state by 13 dpi.

Furthermore, all groups had similar protection levels, which were around 80% for all the influenza A subtypes tested. Altogether, these results showed that the Epimedium koreanum Nakai aqueous extract was strong to inhibit viral replication and promote the survival of mice against lethal infections of different subtypes of influenza A viruses. They suggested that oral administration of this aqueous extract has the potential as an effective herbal remedy for prophylaxis and therapeutic applications in both humans and livestock [111].

In vivo validation of anti-influenza capacity of quercetin and chlorogenic acid was tested by Liu and colleagues. The lung index and survival rate were employed to evaluate the protective effect of the compounds. BALB/c female mice were infected with influenza A H1N1 (A/PR/8/34), and then treated with the compounds in different doses through intragastric administration. The inhibition rates were dose dependent. The survival rate and lung index for quercetin treatment (80%, 0.72 ± 0.07) at 960 mg/kg/day were similar to zanamivir at 480 mg/kg/day (90%, 0.65 ± 0.10) [137]. 

The key notes derived from these in vivo studies are categorized in Table 2.

## 3. Conclusions

Quercetin and its derivatives are flavonol compounds predominantly found in fruits and vegetables, and they have a strong reputation for inflammatory diseases treatment. They have long been recognized for their therapeutic effects and used as traditional medicines in different parts of the world, typically as herbal drinks to treat this and several other types of diseases. They have recently gained increasing interest in different treatment protocols. This review introduces the most recent developments in terms of in vitro, in silico, and in vivo evaluation of quercetin and its derivatives against influenza infection, which may highlight experimental parameters and bioavailability issues with regards to these compounds. 

Although the research outcomes are somewhat contradictory and indeed some information regarding adverse effects may not be disclosed on scientific websites, this natural compound may nevertheless possess great therapeutic value and is worthy of further investigation. DNA barcoding should also be applied in a complementary manner for the accurate identification of plant species with well-characterized chemical properties, thus improving the quality of medicines.

## 4. Future Perspective

Regarding the importance and efficacy of quercetin and other flavonols against influenza virus infection, flavonoids have attracted increasing attention as a potential therapeutic to combat infection with SARS-CoV-2, from human coronavirus family, which has been responsible for a rapid worldwide pandemic starting from 2020. A few in silico and computational studies have evaluated libraries of natural products and phenolic compounds against SARS-CoV-2. As a result, quercetin was found among the top 5 most potent compounds to bind strongly to the Spike Protein (S) receptor of the virus [156] and with even stronger affinity for the COVID-19 main protease active site than hydroxy-chloroquine [157]. In addition, rutin was identified to have the potential to bind to its main protease active site [158] with notable inhibitory activity against SARS-CoV-2 main protease [159]. It should be noted that oral administration of quercetin would not achieve drug contact with viral S protein, because of its glycosylation for biotransformation in the gastrointestinal tract [160]. Therefore, using nasal or throat spray containing quercetin in a suitable form was recommended to deliver proper volume to the virus active sites [161] and be effective in clinical trials [162]. Therefore, the flavonoid quercetin appears to be the most promising natural antiviral compound on the basis of its antiviral activity against influenza A virus and SARS-CoV-2 infections and is a noteworthy candidate for further in vivo studies.

## Figures and Tables

**Figure 1 biomolecules-11-00010-f001:**
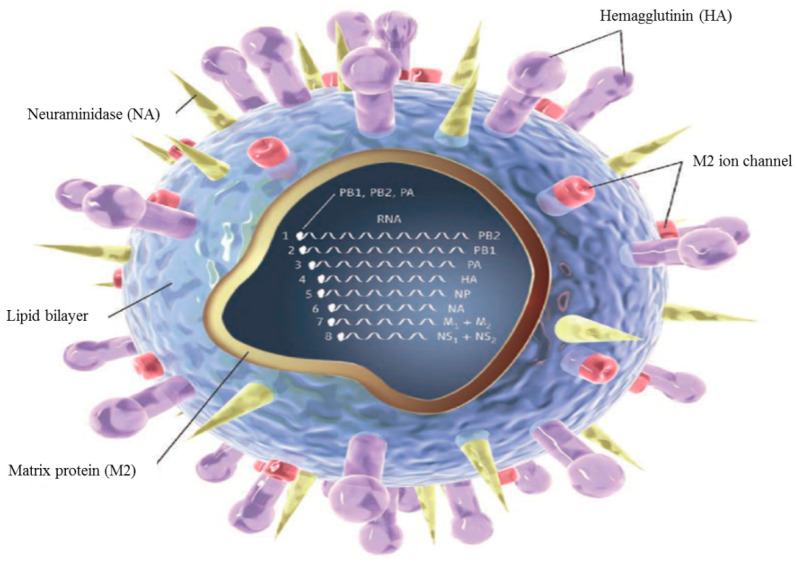
Structure of influenza virus.

**Figure 2 biomolecules-11-00010-f002:**
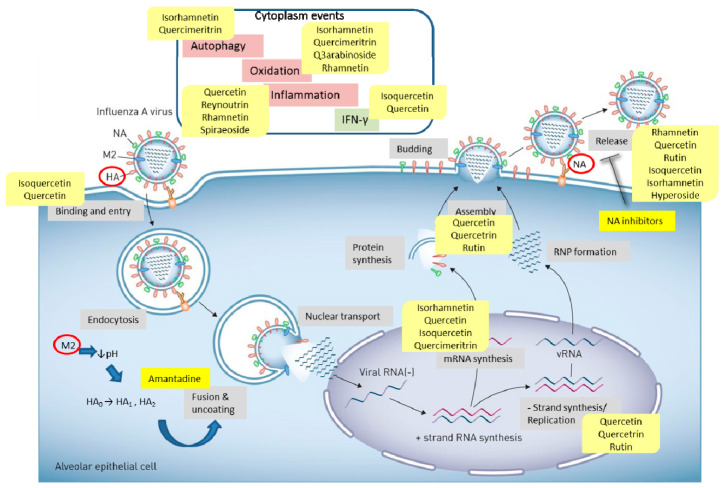
Schematic of the influenza virus life cycle and the pathways affected by quercetin and its derivatives. Influenza virus uses Hemagglutinin (HA) receptor for binding and entry. Membrane 2 (M2) ion channel facilitates fusion and uncoating, and neuraminidase activity (NA) induces the release of progeny virus. Quercetin and its derivatives impact different steps of viral life cycle, such as binding, uncoating, mRNA synthesis, negative-strand synthesis, assembly, and release. They can also affect cytoplasm events to influence the virus indirectly.

**Table 1 biomolecules-11-00010-t001:** Quercetin and its derivatives information (molecular weight and structure).

Common Name (Systematic Name)	Molecular Formula (MW)	Structure
Quercetin (3,5,7,3′,4′- pentahydroxyflavon)	C_15_H_10_O_7_ (302.2)	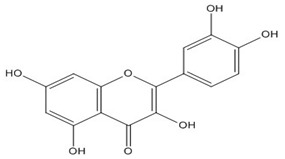
Hyperoside (quercetin- 3-*O*-galactoside)	C_21_H_20_O_12_ (464.3)	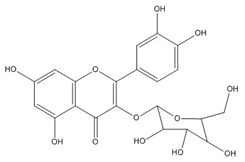
Isoquercetin (quercetin- 3-*O*-glucoside)	C_21_H_20_O_12_ (464.3)	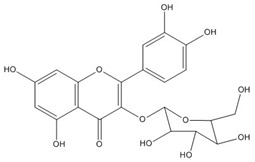
Reynoutrin (quercetin- 3-*O*-xyloside)	C_20_H_18_O_11_ (434.3)	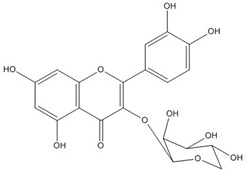
Miquelianin (quercetin- 3-*O*-glucuronide)	C_21_H_18_O_13_ (478.3)	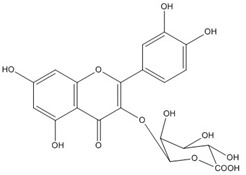
Quercimeritrin (quercetin- 7-*O*-glucoside)	C_21_H_20_O_12_ (464.3)	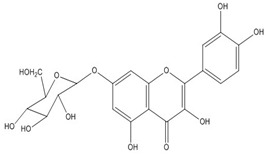
Quercetin 3,4′- diglucoside	C_27_H_30_O_17_ (626.4)	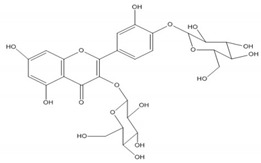
Quercetin-3-glucoside- 7-rhamnoside	C_27_H_30_O_16_ (610.5)	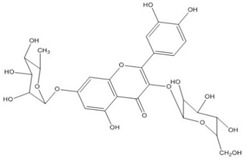
Quercetin 3,7- diglucoside	C_27_H_30_O_17_ (626.5)	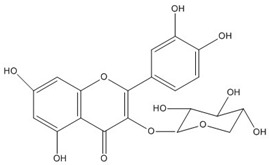
Rutin (quercetin- 3-rutinoside)	C_27_H_30_O_16_ (610.5)	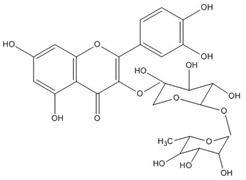
Quercetrin (quercetin 3-*O*-rhamnoside)	C_21_H_20_O_11_ (448.4)	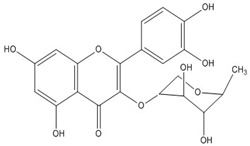
Spiraeoside (quercetin 4-*O*-glucoside)	C_21_H_20_O_12_ (464.3)	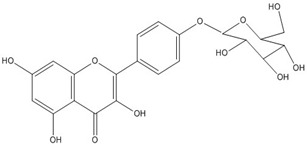
Quercetin 4,7- *O*-diglucoside	C_27_H_30_O_17_ (626.5)	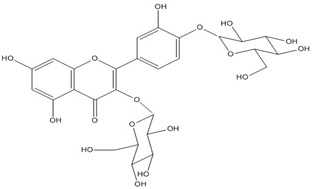
Quercetin 3- *O*-arabinoside	C_20_H_18_O_11_ (434.3)	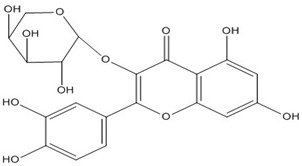
Rhamnetin (quercetin 7-*O*-methyl)	C_16_H_12_O_7_ (316.2)	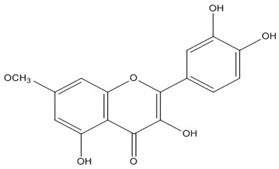
Quercetin 3′- methyl ether	C_16_H_12_O_7_ (316.3)	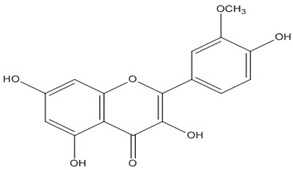
Quercetin-3-*O*-α-*L*-rhamnopyranoside	C_21_H_20_O_11_ (228.4)	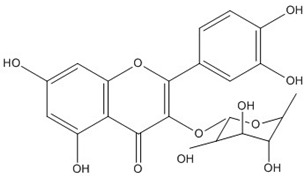

Adapted from Gansukh et al. [75] with modification.

**Table 2 biomolecules-11-00010-t002:** Animal model and human studies of quercetin and derivatives.

Compound Studied	Human/Animal Study	Study	Reference
Aqueous extract of propolis containing rutin and quercetin	In mice infected with A/PR8/34 (H1N1)	Experimental postinfection of the extract, decrease HA titer in the lung suspensions, slight decrease in mortality and increase in survival length	[142]
Quercetin	1 mg/mL administered orally. In adult male Swiss albino mice infected with influenza virus (A/Hong Kong/8/68)	Decreased oxidative stress by decreasing lipid peroxidation, and antioxidant enzymes; superoxide dismutase and catalase	[143]
Quercetin	In mice infected with A/Udorn/317/72 (H3N2) influenza virus	Decreased lipid peroxidation, protected lungs by maintaining the levels of endogenous antioxidant enzymes	[145]
Rutin and quercetin	In mice infected with A/Aichi/2/68 (H3N2)	Quercetin showed stronger activity and caused oxidative damage in healthy mice but antioxidant activity in infected animals	[146]
Combination of selenium, vitamin E, *N*-acetyl-cysteine/glutathione, resveratrol and quercetin	Against H5N1 influenza in humans	Antagonized the pathogenic processes of H5N1 influenza in humans	[149]
Short-term quercetin feeding before viral challenge	In 4-week-old male ICR mice forced to run on a treadmill for 3 consecutive days to fatigue, inoculated with A/PR8/34 (H1N1)	Reduced the susceptibility to infection	[153]
Isoquercetin	In 4-week-old female BALB/c mice infected with human influenza A virus	Decreased IFN-γ, RANTES, and iNOS as indicators of lung inflammation and virus titers	[95]
Quercetin-3-*O*-β-*D*-glucuronide isolated form *Polygonum perfoliatum* L.	In 3 and 6 mg/kg concentrations. In mice infected with influenza virus	It could suppress lung edema caused by virus at 28% and 20%, respectively	[154]
quercetin 3-rhamnoside (Q3R)	In mice infected with A/WS/33 influenza virus	Decreased weight loss, necrosis, inflammation, mortality, and lung virus titer	[155]
*Epimedium koreanum* Nakai aqueous extract containing quercetin	In BALB/c mice infected with 5MLD_50_ of H1N1, H5N2, H7N3, and H9N2 influenza viruses	Reduced viral titers and body weight loss. Inhibited viral replication and promoted the survival	[111]
Quercetin and chlorogenic acid	In BALB/c female mice infected with A/PR8/34 (H1N1)	Improved lung index and survival rate	[137]

## Data Availability

No new data were created or analyzed in this study. Data sharing is not applicable to this article.

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
