# Peer review of "Quercetin as a Natural Therapeutic Candidate for the Treatment of Influenza Virus"

_biomolecules, 2020, doi:10.3390/biom11010010_

Round 1

Reviewer 1 Report

The review of Parvaneh Mehrbod, Dorota Hudy, Divine Shyntum, Jaroslaw Markowski, Marek Los, and Saeid Ghavami is devoted to the anti-influenza effects and mechanisms of action of Quercetin and related flavonoids including in vitro, in vivo, and in silico studies. The possible application for Quercetin and its derivatives, medicinal plant extracts containing this flavonoid also discussed in a complementary therapy and control flu and other respiratory infections including COVID-19 based on the targets. The topic is relevant since a lot of work is being done in this area. The authors have done good work on data summarizing. The manuscript is suitable for publication in Biomolecules after correction and modification (corrections are shown in the text in yellow color including the proposals for corrections, such as pp. 3,5,6,8,9,10,11,13,14, etc.). There are many errors and spellings in the text of the manuscript, no articles in many places, so, it should be corrected with English proofreading too.

Reviewer   

Author Response

The review of Parvaneh Mehrbod, Dorota Hudy, Divine Shyntum, Jaroslaw Markowski, Marek Los, and Saeid Ghavami is devoted to the anti-influenza effects and mechanisms of action of Quercetin and related flavonoids including in vitro, in vivo, and in silico studies. The possible application for Quercetin and its derivatives, medicinal plant extracts containing this flavonoid also discussed in a complementary therapy and control flu and other respiratory infections including COVID-19 based on the targets. The topic is relevant since a lot of work is being done in this area. The authors have done good work on data summarizing. The manuscript is suitable for publication in Biomolecules after correction and modification (corrections are shown in the text in yellow color including the proposals for corrections, such as pp. 3,5,6,8,9,10,11,13,14, etc.). There are many errors and spellings in the text of the manuscript, no articles in many places, so, it should be corrected with English proofreading too.

Answer: We appreciate the careful consideration of the respected reviewer. We have done all the corrections and suggestions and addressed them. We also did some additional corrections in the relevant parts. The manuscript was also edited by a native English speaker and some of the sentences were changed compared to the original writing (certificate provided to editorial office).

P1, Line 2: “as” was added.

P 7, Line 270: “the aglycone form of rutin” was added.

P 8, Line 320: “et al.” was added.

P 9, Line 349: “quercetin derivatives” was added.

P 9, Line 356: related reference was added.

P 9, Line 362: “have stronger inhibitory effects” was added.

P 9, Line 364: “the presence of more” was added

P 9, Line 366: “dimeric” is correct.

P 9, Line 370: “of” was added.

P 9, Line 384: “rumpictuside” was corrected.

P 11, Line 449: related reference was added.

P 11, Line 463-464: “described anti-influenza activity exhibited by a methanol extract of this plant, protecting cell viability and viral titer decrement” was added.

P 12, Line 489: “to counteract” was added.

P 12, Line 494: “the” was added.

P 13, Line 534: related reference was added.

P 13, Line 555: related reference was added.

P 14, Line 561: related reference was added.

P 14, Line 586: “To verify” was added.

P 15, Line 608: “Lee et al. (2016) [130] suggested that” was added.

P 16, Line 657: human was corrected.

P 16, Line 672: “, and oseltamivir” were corrected.

P 17, Line 702: related reference was added.

P 17, Line 704: “hr after virus inoculation” was added.

P 17, Line 723: related reference was added.

P 20, Line 820: “and other flavonols” was added.

P 20, Line 821: “flavonoids have …” was added.

P 20, Lines 823-824: “have evaluated libraries” was added.

P 20, Line 824-825: “As a result, quercetin was found” was added.

P 20, Line 833: “flavonoid quercetin” was added.

P 20, Line 833: “promising natural antiviral compound” was added.

P 20-P21: more abbreviated words were added.

P 25, Line 903: “Adapetd from Gansukh et al. [72] with modification” was added.

Reviewer 2 Report

In this review, the authors have examined a significant sample of the scientific literature on influenza infection and quercetin as a natural product. However, there are a number of omissions and inaccuracies, which I will comment on below.

I will start with the critical notes from the title. What kind of candidate is quercetin against the flu virus? In the presidential election, perhaps? Probably the authors keep in mind that quercetin is a natural candidate for the treatment of influenza viral infection. This is something else entirely.

When a drug is presented to treat a disease, the pathogenesis of that disease must first be studied. In the present work, which is an analysis of literature data on the potential of quercetin to cure us of influenza, I did not see such a description. The pathogenesis of influenza has been studied since the last decade of the last century and is constantly enriched with new additions - lipid peroxidation, oxidative stress, cytokine storm, fight between pro-inflammatory and anti-inflammatory factors, etc. The target organ of the viral replicas. From the data in Figure 2, I have the impression that the influenza virus attacks all the cells encountered.

The facts presented so far will make it possible to substantiate well the current analysis of the scientific literature contained in this review.

A chapter on viral pathogenesis should be added at the beginning of the review. And only then to report the data from research on the role of quercetin.

I recommend the following articles:

  1. Oda, T., Akaike, T., Hamamoto, T., Suzuki, F., Hirano, T., H. Maeda. (1989). Oxygen radicals in influenza-induced pathogenesis and treatment with pyran polymer-conjugated SOD. Science. 244:974-976
  2. Schwarz, K. (1996). Oxidative stress during viral infection: a review. Free Radic Biol Med. 215641-9
  3. Akaike, T., Noguchi, Y., Ijiri, S., Setoguchi, K., Suga, M., Zheng, Y.M., Dietzschold, B., Maeda, H. (1996). Pathogenesis of influenza virus-induced pneumonia: involvement of both nitric oxide and oxygen radicals. Proc Natl Acad Sci USA. 93: 2448-2453
  4. Peterhans, E. (1997). Oxidants and antioxidants in viral diseases: disease mechanisms and metabolic regulation. J Nutr .127: 962S-965S
  5. Peterhans, E. (1997) Reactive oxygen species and nitric oxide in viral diseases. Biol Trace Elem Res. 56: 107-16
  6. Murphy, A., Platts-Mills, T. A, Lobo, M., Hayden, F. (1998). Respiratory nitric oxide levels in experimental human influenza. Chest. 114 (2): 452-456
  7. Han, S N, Meydani, S N, (2000), Antioxidants, cytokines, and influenza infection in aged mice and elderly humans. J. Infect. Dis. 182:S74-S80
  8. Javoby, D. B. (2002). Virus-induced asthma attacks. JAMA 287(6), 755-761
  9. Ji C., Rouzer C.A., Marnett L.J. and Pietenpol J.A. (1998). Induction of cell cycle arrest by the endogenous product of lipid peroxidation, malondialdehyde. Carcinogenesis. 19: 1275-1283
  10. Cox, N., Hughes, J. (1999). New options for the prevention of influenza. N Engl J Med. 341(18): 1387-1388
  11. Kakishita, H., and Hattori, Y. (2001). Vascular smooth muscle cell activation and growth by 4-hydroxynonenal. Life Sci. 69: 689-697
  12. Liu, B., Mori, I., Hossain, M. J., Dong, L., Chen, Z., Kimura, Y. (2003). Local immune response to influenza virus infection in mice with a targeted disruption of perforin gene. Microbiol Pathol. 34:161-167
  13. Love, S., and Wiley C.A. (2002). Viral diseases. In: Greenfield’s neuropathology. Vol. 2. Graham D.I. and Lantos P.L. (eds). Arnold. London, New York, New Delhi, pp 1-105
  14. Mileva, M., Bakalova, R., Tantcheva, L., Galabov, A. S. (2002). Effect of immobilization, cold and cold-restraint stress on liver monooxygenase activity and lipid peroxidation of influenza virus infected mice. Arch of Toxicol. 76:96-103
  15. Mileva, M., Bakalova R., Tantcheva, L., Galabov, A. S., Ribarov, St. (2000). Effect of vitamin E supplementation on lipid peroxidation in blood and lung of influenza virus infected mice. Comp Immunol Microbiol & Infec Dis. 25:1-11
  16. Palamara, A.T., Nencioni, L., Aquilano, K., De Chiara, G., Hernandez, L., Cozzolino, F., Ciriolo, M.R. and Garaci, E., 2005. Inhibition of influenza A virus replication by resveratrol. Journal of Infectious Diseases, 191(10), pp.1719-1729
  17. Saladino, R., Barontini, M., Crucianelli, M., Nencioni, L., Sgarbanti, R. and Palamara, A.T., 2010. Current advances in anti-influenza therapy. Current medicinal chemistry, 17(20), pp.2101-2140.
  18. Valyi-Nagy, T., and T.S. Dermody (2005). Role of oxidative damage in the pathogenesis of viral infections of the nervous system. Histol Histopathol. 20:957-967
  19. Vlahos, R., Stambas, J., Selemidis, S. (2012). Suppressing production of reactive oxygen species (ROS) for influenza A virus therapy. Trends Pharmacol Sci. 33(1):3
  20. Mileva М. (2016) Oxidative Stress as a Target for Medication of Influenza Virus Infection. Acta Microbiol. Bulg., 3-9

Most of the articles discussed in the chapter “In vitro and in silico studies against influenza virus”

concerning observations on polyphenol-rich plant extracts. The controlled viral cytopathic effect observed in MDS cells does not yet mean viral protection in the infected body. Moreover, alcoholic extracts contain a rich set of biologically active substances that have a polyphenolic structure. In this case, it is more correct to talk about polyphenol-rich extracts containing quercetin.

For a more complete clarification of Figure 2, I recommend analyzing the fact that the pathways by which quercetin can cross the cell membrane.

  1. Pawlikowska-Pawlęga, Bożena, Wiesław Ignacy Gruszecki, Lucjan Misiak, Roman Paduch, Tomasz Piersiak, Barbara Zarzyka, Jarosław Pawelec, and Antoni Gawron. "Modification of membranes by quercetin, a naturally occurring flavonoid, via its incorporation in the polar head group." Biochimica et Biophysica Acta (BBA)-Biomembranes 1768, no. 9 (2007): 2195-2204.

Author Response

Reviewer #2:

  • Probably the authors keep in mind that quercetin is a natural candidate for the treatment of influenza viral infection. This is something else entirely.

Answer: It was a very thoughtful feedback. We changed the title was changed to “Quercetin as a Natural Therapeutic Candidate for the Treatment of Influenza Virus”.

  • A chapter on viral pathogenesis should be added at the beginning of the review. And only then to report the data from research on the role of quercetin.

Answer: We added a chapter on viral pathogenesis “Introduction”, before “Medicinal plants”. P 3-6, Lines 91-237.

  • Most of the articles discussed in the chapter “In vitro and in silico studies against influenza virus” concerning observations on polyphenol-rich plant extracts. The controlled viral cytopathic effect observed in MDS cells does not yet mean viral protection in the infected body. Moreover, alcoholic extracts contain a rich set of biologically active substances that have a polyphenolic structure. In this case, it is more correct to talk about polyphenol-rich extracts containing quercetin.

Answer: We applied the corrections to the related parts of “In vitro and in silico studies against influenza virus”; P 8, Lines 314, 320, 331

  • For a more complete clarification of Figure 2, I recommend analyzing the fact that the pathways by which quercetin can cross the cell membrane.
  1. Pawlikowska-Pawlęga, Bożena, Wiesław Ignacy Gruszecki, Lucjan Misiak, Roman Paduch, Tomasz Piersiak, Barbara Zarzyka, Jarosław Pawelec, and Antoni Gawron. "Modification of membranes by quercetin, a naturally occurring flavonoid, via its incorporation in the polar head group." Biochimica et Biophysica Acta (BBA)-Biomembranes 1768, no. 9 (2007): 2195-2204.

Answer: We added the related information under the section” Quercetin and its derivatives”

It is worth noting that Pawlikowska-Pawlęga et al. (2007), [87] in their study highlighted modifications in the phospholipid membranes of human skin fibroblasts (HSF) after exposure to quercetin, which was found to localize in the polar head regions of the membrane. It could be detected on membranes in the cytoplasm, around the nuclear envelope and inside the nucleus and nuclei by increasing the exposure time. This property affects its pharmacological activities quality. Figure 2 shows the schematic mechanisms of quercetin and its derivatives on the targeting of  influenza virus life cycle in different parts of the cell. P 7, Lines 267-282.

Round 2

Reviewer 2 Report

See the file below.
